# Lipoteichoic Acid Accelerates Bone Healing by Enhancing Osteoblast Differentiation and Inhibiting Osteoclast Activation in a Mouse Model of Femoral Defects

**DOI:** 10.3390/ijms21155550

**Published:** 2020-08-03

**Authors:** Chih-Chien Hu, Chih-Hsiang Chang, Yi-min Hsiao, Yuhan Chang, Ying-Yu Wu, Steve W. N. Ueng, Mei-Feng Chen

**Affiliations:** 1Bone and Joint Research Center, Chang Gung Memorial Hospital, Taoyuan 33305, Taiwan; r52906154@cgmh.org.tw (C.-C.H.); ccc0810.chang@gmail.com (C.-H.C.); skyyimin@gmail.com (Y.-m.H.); yhchang@cgmh.org.tw (Y.C.); l43219713@hotmail.com (Y.-Y.W.); wenneng@cgmh.org.tw (S.W.N.U.); 2Department of Orthopedic Surgery, Chang Gung Memorial Hospital, Taoyuan 33305, Taiwan; 3College of Medicine, Chang Gung University, Taoyuan 33302, Taiwan; 4Graduate Institute of Clinical Medical Sciences, College of Medicine, Chang Gung University, Taoyuan 33302, Taiwan

**Keywords:** lipoteichoic acid, bone healing, femoral defect, fracture, alkaline phosphatase, osteopontin, endochondral ossification, osteoblast, osteoclast, fluorochrome dynamic labeling

## Abstract

Lipoteichoic acid (LTA) is a cell wall component of Gram-positive bacteria. Limited data suggest that LTA is beneficial for bone regeneration in vitro. Thus, we used a mouse model of femoral defects to explore the effects of LTA on bone healing in vivo. Micro-computed tomography analysis and double-fluorochrome labeling were utilized to examine whether LTA can accelerate dynamic bone formation in vivo. The effects of LTA on osteoblastogenesis and osteoclastogenesis were also studied in vitro. LTA treatment induced prompt bone bridge formation, rapid endochondral ossification, and accelerated healing of fractures in mice with femoral bone defects. In vitro, LTA directly enhanced indicators of osteogenic factor-induced MC3T3-E1 cell differentiation, including alkaline phosphatase activity, calcium deposition and osteopontin expression. LTA also inhibited osteoclast activation induced by receptor activator of nuclear factor-kappa B ligand. We identified six molecules that may be associated with LTA-accelerated bone healing: monocyte chemoattractant protein 1, chemokine (C-X-C motif) ligand 1, cystatin C, growth/differentiation factor 15, endostatin and neutrophil gelatinase-associated lipocalin. Finally, double-fluorochrome, dynamic-labeling data indicated that LTA significantly enhanced bone-formation rates in vivo. In conclusion, our findings suggest that LTA has promising bone-regeneration properties.

## 1. Introduction

Delayed bone healing leads to a diminished quality of life, a personal financial burden and increased medical care costs. Thus, novel therapies to enhance bone healing are sorely needed. The U.S. Food and Drug Administration defines a non-union as a fracture that is at least nine months old, which has not shown any sign of healing for three consecutive months [1]. Clinically, non-unions are classified according to their morphology and radiographic appearance in atrophic and hypertrophic non-unions [2,3].

Current state-of-the-art therapeutic methods for impaired bone healing can be divided into four categories: conservative therapy, surgical therapy, molecular targets for local or systemic applications and novel bone substitute grafts. Conservative therapeutic methods include dynamization of intramedullary fixation, increased weight bearing, low-intensity pulsed ultrasound or extracorporeal shock wave therapy [4,5]. Surgical-therapy strategies involve debridement of sclerotic edges combined with autogenous or allogenic bone grafts [6]. Harvesting and transplanting autogenous bone from the iliac crest is the gold standard for supporting bone healing in non-unions. However, limited autografts are available and are accompanied with high risks during harvest for wound infection and postoperative pain [7]. Allogenic bone grafts provide a relatively safe alternative over autografts, which have donor-site disadvantages due to their processing in terms of sterilization and storage, which in turn causes losses of osteogenic and osteoinductive capabilities [8]. The molecular targets for local or systemic applications (parathyroid hormone, the Wnt-signaling pathway, sclerostin, bone morphogenic proteins, platelet-derived growth factor and fibroblast growth factor) have provided benefits for regulating bone regeneration [9]. Novel bone substitute grafts are currently being investigated, some of which are already being applied clinically, such as calcium-based bone grafts (calcium phosphate ceramics, calcium phosphate cements and calcium sulfate), bioactive glass, organic bone grafts and three-dimensional [3D] printed scaffolds [10,11,12]. Despite tremendous scientific and clinical efforts, impaired bone healing still represents a challenging complication following a bone injury. Thus, new strategies to enhance bone healing are warranted.

Lipoteichoic acid (LTA) is a major component of Gram-positive (GP) bacterial cell walls. LTA binds specifically to target cells via CD14 and Toll-like receptor 2 (TLR2), or non-specifically through membrane phospholipids. Previous data show that LTA can activate immune responses through down-stream signaling pathways, including nuclear factor kappa-light-chain-enhancer of activated B cells (NF-κB) expression, phosphoinositide 3-kinase activation and mitogen-activated protein kinase (MAPK) activation. To date, only a few reports have described the impact of LTA on bone formation. LTA-stimulated mesenchymal stem cells showed significantly upregulated expression of osteogenesis markers, such as alkaline phosphatase (ALP), calcium deposition, type-I collagen and Runx2 [13]. We recently found that periprosthetic joint infection (PJI) due to GP or Gram-negative (GN) bacteria can lead to different rates of aseptic loosening after reimplantation [14]. PJI caused by GN bacteria indicates a higher risk of aseptic loosening after reimplantation, although this phenomenon did not occur in PJI caused by GP bacteria. Additionally, exposure to GP bacteria-derived LTA inhibits the differentiation of bone marrow-derived macrophages (BMMs) to osteoclasts (Ocls) [15]. LTA inhibited receptor activator of the nuclear factor-kappa B ligand (RANKL)-induced expression of c-Fos and nuclear factor of activated T cells 1 in a TLR2-dependent manner [15]. LTA also sustained the phagocytic capacity of BMMs even after Ocl differentiation [15]. To further clarify whether LTA improves the process of bone healing, we designed the current study with three specific aims. First, we generated a bone-defect mouse model to examine whether LTA treatment can accelerate bone healing. Second, we conducted in vitro experiments to examine the differential effects of LTA on osteoblastogenesis and osteoclastogenesis. Finally, we used a double-fluorochrome labeling model to observe whether LTA may accelerate the onset of osteogenesis in mice with femoral defects in vivo. We found that LTA treatment induced bone bridge formation, endochondral ossification and healing of fractures in mice with femoral bone defects. We also found that LTA directly enhanced indicators of osteogenic factor-induced MC3T3-E1 cell differentiation. In addition, we observed that LTA inhibited NF-κB-induced Ocl activation in vitro and identified six molecules that may be related to LTA-induced bone healing. Our findings suggest that LTA is a promising molecule for promoting immuno-modulatory bone regeneration in bone biomaterials.

## 2. Results

### 2.1. LTA Accelerated Bone Healing in Mice with Femoral Bone Defects

Micro-computed tomography (microCT) images were used to reconstruct the femoral morphology and analyze the bone density. In the sham-treated group of femoral-defect mice, significant gaps were still observed on Day 7 after surgery, and cancellous bone formation was seen on Day 14 (Figure 1a). In the LTA-treated mice, significant bone formation was observed on Day 7 after surgery, and condensed bones were seen on Day 14. Body-weight recovery occurred earlier in the LTA-treated group than in the sham group (Figure 1b). Higher bone density-related parameters were observed in the LTA-treated group than in the sham control group, in terms of the trabecular thickness, trabecular number, trabecular spacing, bone volume/tissue volume fraction (BV/TV) and bone surface/TV (BS/TV) density (Figure 1c). Treatment with LTA, but not PBS, increased morphometric bone indices.

Hematoxylin and eosin (H&E) staining revealed an early-onset dense bone bridge by Day 7 in LTA-treated mice, whereas mice in the sham group showed hematoma formation (Figure 2a). Safranin-O and Masson’s trichrome staining images were acquired. Many undifferentiated mesenchymal cells were present in the callus and in the areas of inflammation. In the sham control group, we observed the bone bridge, hypertrophic chondrocytes, proliferating chondrocytes and osteoblasts (Obs) on Day 14 after introducing the femoral bone defects. In the LTA group, the bone bridge appeared earlier (Day 7) after introducing the femoral defects; moreover, mature osteocytes (Ocys) and Ocls were present in trabecular bone on Day 14. These results demonstrate that endochondral ossification was nearly complete on Day 14 in the LTA group, but not in the sham group (Figure 2a). Immunofluorescence analysis showed intense osterix and cathepsin K staining surrounding the trabecular bones in LTA-treated group, whereas diffuse staining was observed in the sham control group (Figure 2b). We also quantified the intensities of osterix and cathepsin K signals relative to that of DAPI (bone tissue containing area). Bone remodeling occurred earlier in the LTA group (Appendix A).

### 2.2. LTA Had Beneficial Effects on Bone Formation In Vitro

LTA directly enhanced osteogenic factor-induced MC3T3-E1 cell differentiation, including the ALP levels (Figure 3a,b) and calcium deposition (Figure 3c,d). LTA also increased osteopontin secretion during MC3T3-E1 cell differentiation (Figure 3e). During RANKL-induced Ocl activation, LTA inhibited the activity of the Ocl-associated protein, tartrate-resistant acid phosphatase (TRAP; Figure 3f,g). However, LTA did not alter the RANKL-enhanced expression of cathepsin K at either the protein- or mRNA-expression levels (Figure 3h–j).

### 2.3. LTA Positively Regulated Bone Healing by Promoting Protein Secretion from Obs

Antibody arrays enable high-throughput analysis of protein-expression levels on a large scale. To examine the possible molecules involved in LTA-enhanced osteogenesis, an antibody array was used. Culture supernatants were collected (with or without LTA pretreatment) and analyzed by performing protein-array assays. Each spot was measured based on changes in the signal intensity to identify potential secretory factors induced by LTA, resulting in enhanced bone formation (Figure 4a). The intensity of all spots was evaluated with respect to those of the references and negative spots. We found that six of the 111 spots showed changes between different groups and these spots were selected for further intensity measurements (Figure 4b). The expression levels of all the six proteins were higher in the LTA group than in the control group. These six proteins were monocyte chemoattractant protein 1 (MCP-1), chemokine (C-X-C motif) ligand 1 (CXCL1), cystatin C, growth/differentiation factor 15 (GDF15), endostatin and neutrophil gelatinase-associated lipocalin (NGAL), which may be related to LTA-accelerated bone healing during bone remodeling.

### 2.4. LTA Enhanced Dynamic Bone-Formation Rates In Vivo

To examine the impact of LTA on the dynamic bone-formation rate in vivo, we performed a double-labeling study of the mouse femoral defects, using xylene orange (XO) and calcein green (CG). Newly formed CG-labeled bone cells surrounded the trabecular surfaces in the sham group, whereas these cells were present inside the trabecular bones in the LTA group. Histomorphometric analysis revealed that static bone-formation parameters were significantly different between the LTA and sham groups. A fully connected bone bridge formed in the LTA group, but not in the sham group (Figure 5). These results demonstrate that the dynamic bone-formation rates in the LTA group were higher than those in the sham group. Furthermore, our results show a higher rate of trabecular remodeling in the LTA group, compared with sham group. In general, labeling with XO quickly faded in the LTA group, but clear labeling was observed in the trabecular bone areas in the sham group. Calcein green labeling was found on the trabecular surface in the sham group, whereas CG signaling was present inside the trabecular region in the LTA group. These results show the potential for LTA to enhance endochondral ossification in mice with femoral bone defects.

## 3. Discussion

LTA is a cell wall component of GP bacteria that is recognized by toll-like receptor 2 on the cell surface, which then initiates signaling cascades including the NF-κB and MAPK pathways [16]. LTA has been isolated from different species of GP bacteria, and it has various chemical structures and functional activities [17]. Our recent data indicate that PJI caused by GN bacteria indicates a higher risk of aseptic loosening after reimplantation, mainly because of lipopolysaccharide (LPS)-mediated effects on Ocl differentiation [14]. Interestingly, LPS (but not LTA) reduced both the number of trabeculae and the bone-mineral density in mice. Because LTA does not cause aseptic loosening, we were curious about its potential effects on bone healing. In this study, we demonstrated that LTA isolated from *Staphylococcus aureus* accelerated bone healing in vivo. In our in vitro studies, LTA directly enhanced Ob differentiation and inhibited Ocl activation in MC3T3-E1 and RAW264.7 cell models, respectively. Moreover, LTA stimulated Obs to secrete several proteins, including MCP-1, CXCL1, cystatin C, GDF15, endostatin and NGAL, which may provide a regulatory function through an autocrine mechanism. It was previously reported that the MCP-1 production by Obs in bone specimens from patients with *S. aureus*-associated osteomyelitis can stimulate the proliferation of osteoblastic cells [18]. It was also shown that Obs could release CXCL1, which attracted Ocl precursors to the bone environment [19]. Cystatin C is synthesized by bone cells and affects bone morphogenetic protein-signaling cascades in osteoblastic cells and then promotes Ob differentiation, mineralization and bone formation [20,21,22]. GDF-15, endostatin and NGAL were previously found to inhibit RANKL-induced Ocl formation [23,24,25,26]. Endostatin attenuated vascular endothelial growth factor-A-induced osteoclastic bone resorption [27]. Based on these findings, we speculate that LTA inhibited Ocl activation and bone resorption through LTA-induced expression of MCP-1, CXCL1, cystatin C, GDF15, endostatin and NGAL. These molecules may be related to LTA-accelerated bone healing during bone remodeling.

Immune responses and circulating immunity-related factors regulate skeletal cells during the processes of normal and pathological bone formation. Although bacterial factors can trigger immunity that induces pro-osteogenic pathways, these usually pale in significance due to osteolysis and concerns of systemic inflammation. Limited data have been published demonstrating that LTA may affect Obs and Ocls in vitro. Staphylococcal LTA inhibited the phosphorylation of extracellular signal-regulated kinase and c-Jun N-terminal kinase in Ocl precursors, which were treated with macrophage colony-stimulating factor and RANKL, concomitantly with a decreased DNA-binding activity of activator protein 1 [28]. The LTA molecule of *Enterococcus faecalis* is an Ocl inhibitor that significantly inhibited osteoclastogenesis of BMMs in the presence of RANKL [29]. In contrast, the mRNA- and protein-expression levels of osteogenesis markers were significantly upregulated after treatment with the staphylococcal LTA; enhanced ALP positivity was found in the LTA groups; and calcium nodule formation increased simultaneously [13]. LTA may function as an osteo-stimulatory factor through a synergy with osteoinductive signals [17]. Although these studies mentioned above indicate the impacts of LTA on Obs and Ocls in vitro, no reports have described the effects of LTA on bone healing in vivo. In this study, we demonstrated that LTA not only accelerated bone healing, but also enhanced dynamic bone formation.

Bone lining tissues contain residual osteal-macrophages called osteomacs, which interact with OBs and are located immediately adjacent to OBs; they regulate bone formation and play an important role in regulating the bone healing process [30]. The F4/80(+) Mac-2(−/low) TRACP(-) osteomacs are present within the bone injury site and persisted throughout the healing duration [31]. Osteomacs are required for deposition of collagen type 1 matrix and bone mineralization in a mouse model of tibial injury [31]. Additionally, induction of M2 macrophages through interleukin 4 and 13 significantly enhanced bone formation during fracture healing [32]. Depletion of M1 macrophages was also found to reduce callus properties, alter the cytokine expression profiles during early bone repair and impair the bone healing process [33]. These studies strongly indicate that macrophages influence bone healing.

Osteointegration refers to a process whereby bone cells come into direct contact with an orthopedic implant. Osteointegration includes mesenchymal stem cell attachment, proliferation, and differentiation into Obs on the implant surface, resulting in the formation of mineralized bone around the implant [34,35]. Because of the demand for artificial joint-replacement surgery and the common use of metal implants in orthopedic surgery, osteointegration is becoming a crucial research topic. Approximately 30,000 patients require joint-replacement surgery in Taiwan per year. For successful total-joint arthroplasty, the long-term fixation of implants (due to osteointegration) is crucial. Additionally, to improve the efficacy of other orthopedic surgical procedures, the metal implants used in many orthopedic treatments (such as fracture surgery, rod implantation and chiropractic therapy) require osteointegration. For example, posttraumatic bone fractures are usually fixed with implanted devices to correct the position of bone fragments and to assist in the healing process. Therefore, applying LTA on the surface of an orthopedic implant to improve osteointegration may represent a novel and viable treatment strategy. Moreover, new strategies to enhance bone healing are still needed. We believe that LTA may contribute to the process of bone healing. Using LTA to promote bone healing has the following advantages. First, because LTA comes from a GP bacterial cell wall, it is easy to obtain. A supplier would only need to culture a large number of bacteria and collect LTA from the cell wall. Consequently, the manufacturing cost should be relatively low. Finally, if LTA is used as a clinical drug, it should be easy for suppliers to provide LTA at an appropriate dosage and purity [36]. In conclusion, our findings suggest that LTA shows promise as an immuno-modulatory biomaterial for bone regeneration.

## 4. Materials and Methods

### 4.1. Experimental Animal Study

All animal procedures complied with the National Institute of Health guidelines and were reviewed and approved by the local Hospital Animal Care and Use Committee. Initially, 10-week-old male C57BL/6 mice were anesthetized via intraperitoneal injection (0.01 mL/kg body weight) of a 1:1 (vol/vol) mixture of tiletamine-zolazepam (Zoletil; Virbac, Carros, France) and xylazine hydrochloride (Bayer HealthCare AG, Leverkusen, Germany), and the surgical site was shaved and disinfected with povidone-iodine. An incision was made in the skin overlying the right knee joint. A medial parapatellar arthrotomy (with lateral displacement of the quadriceps–patella complex) was performed to access the distal femur. After locating the femoral intercondylar notch, the femoral intramedullary canal was manually pierced with a 25-gauge needle and intrafemorally injected with 10 mg/kg LTA (from *S. aureus*; Sigma-Aldrich, St. Louis, MI, USA) in phosphate-buffered saline (PBS; 10 μL). A stainless-steel rod (with a length of 0.9 mm and diameter of 0.4 mm) was surgically placed in a retrograde manner. A 1-mm defect was formed at the midshaft of the right femur using drill bits of different sizes. The quadriceps–patellar complex was repositioned to the midline, and the surgical site was closed with subcutaneous 6-0 Dexon sutures. Buprenorphine (0.2 mg/kg) was administered subcutaneously every 24 h as an analgesic throughout the experimental duration. The mice were sacrificed on Day 7 or 14 post-surgery. The femur was immediately fixed in formaldehyde (10%) and subjected to micro-CT analysis.

### 4.2. Micro-CT Bone Imaging

Nondestructive ultrastructural analysis was performed using a SkyScan 1176 micro-CT scanner (Bruker Microct, Kontich, Belgium). The samples were wrapped in saline-soaked gauze and scanned at 50 kV with a 0.5-mm aluminum filter. Images with a resolution of 9 μm were reconstructed using GUP-NRecon software (version 1.7.4.2) and analyzed using CTAn software (version 1.15.4.0, SkyScan). The grayscale was based on Hounsfield units, and validated calcium standards were scanned as a density reference (0.25 and 0.75 g/cm^3^ Hydroxyapatite Phantoms). The BV, TV and BV/TV ratio (expressed as a percentage) were calculated for the sections of interest (*n* = 117), which were located between 1 mm above and below the damaged site, comprising all the woven callus and remodeled bone. A 3D image was constructed using CTVox software (version 3.3.0, SkyScan) for illustration purposes.

### 4.3. Histochemistry and Immunofluorescence Staining

The femur samples were incubated in a rapid decalcifier solution, trimmed and paraffin embedded. Four-micrometer-thick sections were subsequently stained with: (1) H&E; (2) safranin-O; (3) Masson’s trichrome stain; (4) an anti-osterix antibody (1:100, ab19027, Abcam; and (5) an anti-cathepsin K antibody (1:100, ab19027, Abcam). The samples were subsequently incubated with secondary Alexa Fluor 488-conjugated anti-rabbit IgG (1:200, A21206, Invitrogen, Carlsbad, CA, USA) for 60 min at 25 °C. The whole slides were digitalized using a NanoZoomer S360 digital slide scanner (Hamamatsu Photonics, Hamamatsu, Japan). Each fluorescence image was then acquired under a fluorescence microscope (DFC7000 T, Leica Microsystems, Wetzlar, Germany).

### 4.4. Ocl and Ob Differentiation

RAW264.7 cells were plated on a 3-well-chamber slide (Nunc Lab-Tek, ThermoFisher Scientific, Waltham, MA, USA; density: 1 × 10^4^ cells/well) and maintained in alpha-modified Eagle’s medium supplemented with 10% fetal bovine serum (FBS) and antibiotics. The cells were differentiated for 5 days into mature Ocls in the presence of LTA (100 ng/mL) or recombinant RANKL (50 ng/mL) alone, or in combination. The formation of mature Ocls was assessed by immunostaining for cathepsin K (ab19027, Abcam) and TRAP (Takara, Shiga, Japan). F-actin was stained with Alexa Fluor 647-conjugated phalloidin (A22287, Invitrogen), whereas DAPI staining was used for nuclear staining (D1306, Invitrogen).

MC3T3-E1 cells (CRL-2593, American Type Culture Collection, Manassas, VA, USA) were plated in a 12-well plate (density: 5 × 10^3^ cells/well) and then cultured in 10% FBS-differentiation medium with or without osteogenic factors (OS). The OS was composed of 5 mM glycerol 2-phosphate, 0.1 µM dexamethasone, and 50 mM ascorbic acid. The cells were treated with or without 100 ng/mL LTA in the osteogenic differentiation medium. Calcification of MC3T3-E1 cells was assessed via Alizarin Red S staining (ScienCell, Carlsbad, CA, USA) to monitor matrix mineralization. Calcium assays were performed using a Calcium LiquiColor Assay (Stanbio laboratory, Boerne, TX, USA) in accordance with the manufacturer’s instructions. A Fast Violet B Salt capsule (catalog number 851-10 CAP, Sigma-Aldrich) was dissolved in naphthol AS-MX phosphate alkaline solution (catalog number 855, Sigma-Aldrich) and then used for ALP staining. Sigma 104^®^ phosphatase substrate (catalog number 104105, Sigma-Aldrich) was used for ALP assays [37].

### 4.5. Enzyme-Linked Immunosorbent Assay (ELISA)

Osteopontin levels were measured using a Mouse Osteopontin DuoSet ELISA Kit (DY441, R&D Systems, Minneapolis, MA, USA), according to the manufacturer’s protocol. Thresholds for intra- and inter-assay coefficients of variation were set at <15%.

### 4.6. Protein Array

We used MC3T3-E1 cell culture medium to survey factors with protein array panels containing 111 well-categorized monoclonal antibodies to compare LTA-specific protein-expression patterns in two groups (with and without LTA). Proteome Profiler Antibody arrays (catalog number ARY028; R&D Systems) were used to simultaneously evaluate the expression levels of multiple factors, including cytokines, chemokines and soluble receptors in the cell culture medium of MC3T3-E1 cells. The signal intensities of spots in the protein array were quantified using ImageJ software (National Institutes of Health, Bethesda, MD, USA).

### 4.7. Fluorochrome Labeling to Measure Dynamic Bone-Formation Rates

Double-fluorochrome in vivo labeling was performed using XO and CG, as described [38]. After introducing femoral bone defects, the mice were subcutaneously injected with XO (80 mg/kg body weight, Day 7) and CG (10 mg/kg body weight, Day 12), after which they were sacrificed on Day 14. The incorporation of fluorochromes into the undecalcified bone sections on glass slides was examined by fluorescence microscopy on methyl methacrylate-embedded sections (40 µm thickness).

### 4.8. Statistical Analysis

All data were obtained from at least three independent experiments. Quantitative data were analyzed with two-way analysis of variance (ANOVA), followed by Bonferroni’s post-hoc test, and are presented as the mean ± standard error of the mean. Body weights were analyzed using two-way repeated-measures ANOVA, followed by Tukey’s post-hoc test. Categorical variables were examined with GraphPad Prism software, version 7.0 (GraphPad Inc., San Diego, CA, USA). Two-tailed *p* values <0.05 were considered to reflect statistically significant differences.

## Figures and Tables

**Figure 1 ijms-21-05550-f001:**
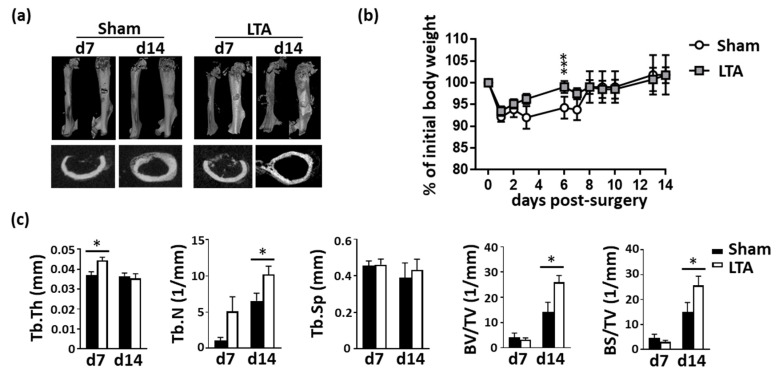
Lipoteichoic acid (LTA) accelerated new bone formation and bone healing in mice with femoral bone defects. (**a**) Mice with femoral bone defects were subjected to intrafemoral injection of phosphate-buffered saline (PBS; sham vehicle control) or LTA. The results of micro-computed tomography (micro-CT) revealed that treatment with LTA (but not the vehicle control) enhanced the bone healing. (**b**) Body weights were measured daily in mice treated with PBS or LTA. All the results were normalized to the initial weight of each mouse. (**c**) Quantitative results of micro-CT analysis in mice treated with PBS (*n* = 5) or LTA (*n* = 6). Data are presented as the mean ± standard error of the mean. Analyses were conducted with two-way analysis of variance followed by Bonferroni’s post-hoc test. ** p* < 0.05, *** *p* < 0.001. Abbreviations: d, day; Tb.Th, trabecular thickness; Tb.N, trabecular number; Tb.Sp, trabecular spacing; BV, bone volume; TV, tissue volume; BS, bone surface.

**Figure 2 ijms-21-05550-f002:**
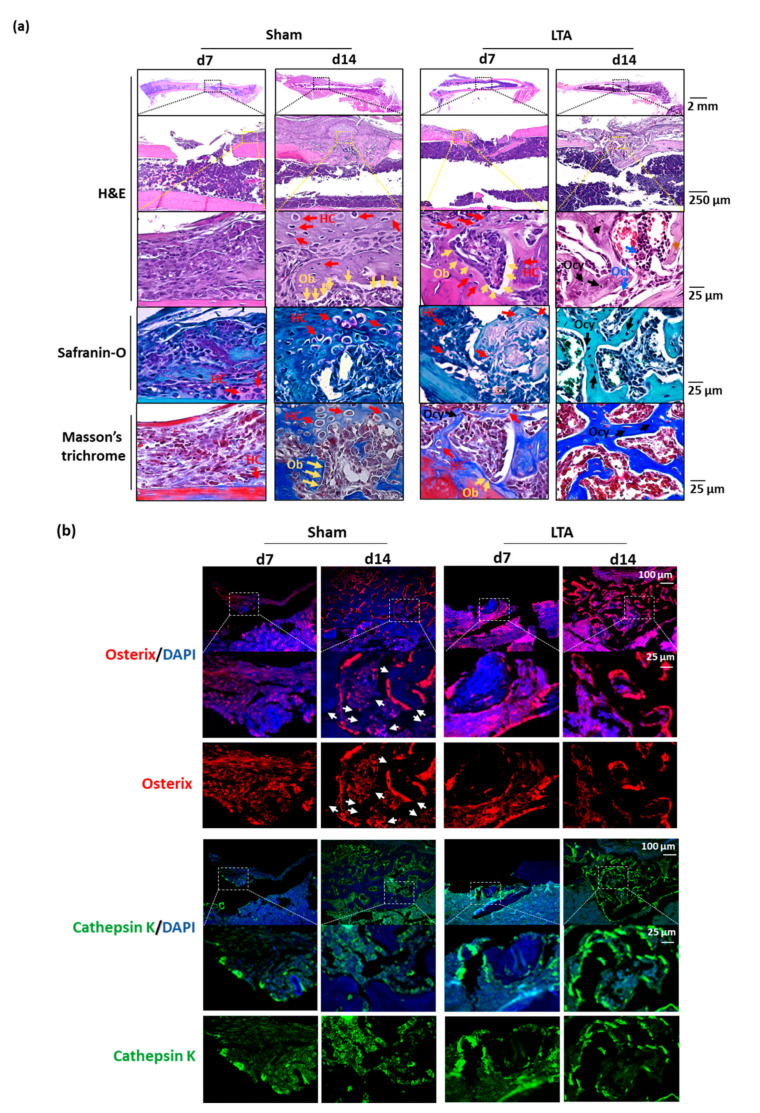
Lipoteichoic acid (LTA) exerts beneficial effects on bone morphology, osterix expression and cathepsin K expression in mice with femoral bone defects. (**a**) Hematoxylin and eosin (H&E), safranin-O and Masson’s trichrome staining revealed the presence of multiple cell types during the bone-regeneration process in the longitudinal histological sections. The bone bridge, hypertrophic chondrocytes (HCs), and osteoblasts (Obs) were observed on Day 14 after femoral bone defects were introduced in the sham control group. The bone bridge, HCs and Obs appeared earlier in the LTA group, on Day 7 after femoral defects were introduced. An extensive network of primary bone (osteocytes, Ocys) and osteoclasts (Ocls) had formed and woven bone was observed on Day 14 following the introduction of femoral defect in the LTA group. (**b**) Immunofluorescence was used to detect osterix (an Ob marker) and cathepsin K (an Ocl marker). Intense osterix and cathepsin K signals were observed surrounding the trabecular bones in the LTA-treated group, whereas the signals for these two proteins were diffusely distributed in the PBS group. The white arrows indicate that fewer newly formed bones were present in the sham group, whereas vigorous bone-formation activity was observed in the LTA group.

**Figure 3 ijms-21-05550-f003:**
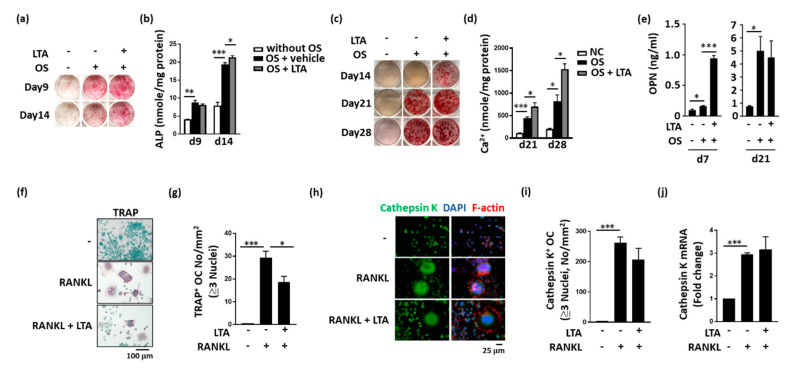
Lipoteichoic acid (LTA) had beneficial effects on bone formation in vitro. (**a**–**d**) LTA enhanced alkaline phosphatase (ALP) expression and calcium levels during osteoblast (Ob) differentiation. (**e**) The osteopontin (OPN) concentrations in the cell culture medium were measured by performing enzyme-linked immunosorbent assays at the indicated time points. OPN-expression levels in the LTA-treated group were higher than those in the control group. (**f**) LTA decreased the differentiation of RAW264.7 cells into tartrate-resistant acid phosphatase (TRAP)-positive osteoclast (Ocl)-like cells. (**g**) Quantitative analysis confirmed that LTA treatment decreased receptor activator of the nuclear factor-kappa B ligand (RANKL)-induced differentiation of TRAP-positive Ocl-like cells. (**h**,**i**) Immunofluorescence staining confirmed that LTA did not affect RANKL-induced differentiation of cathepsin K-positive Ocl-like cells. (**j**) Quantitative polymerase chain reaction analysis was performed to measure the mRNA-expression levels of cathepsin K. The data are presented as the mean ± standard error of the mean. Analyses were conducted with two-way ANOVA, followed by Bonferroni’s post-hoc test. ** p* < 0.05, *** p* < 0.01, *** *p* < 0.001.

**Figure 4 ijms-21-05550-f004:**
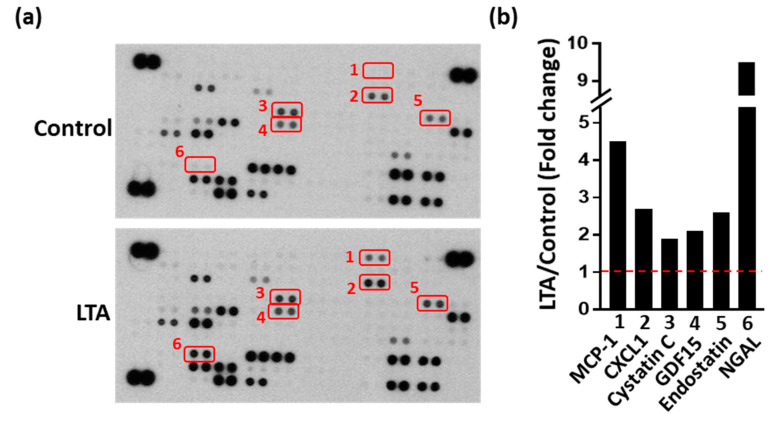
Different protein expression patterns in the culture medium from osteoblastic cells from different groups, as determined by protein-array measurements. (**a**) The expression levels of cytokines, chemokines and soluble factors were measured for each group using protein-array assays. (**b**) Spots with high-intensity changes were measured with the Image J software. The fold-change reflects the expression observed in the LTA group divided by that in the control group. Abbreviations: MCP-1, monocyte chemoattractant protein 1; CXCL1, chemokine (C-X-C motif) ligand 1; GDF15, growth differentiation factor-15; NGAL, neutrophil gelatinase-associated lipocalin. Red box represents two repetitions. Red line represents the basal level (control group).

**Figure 5 ijms-21-05550-f005:**
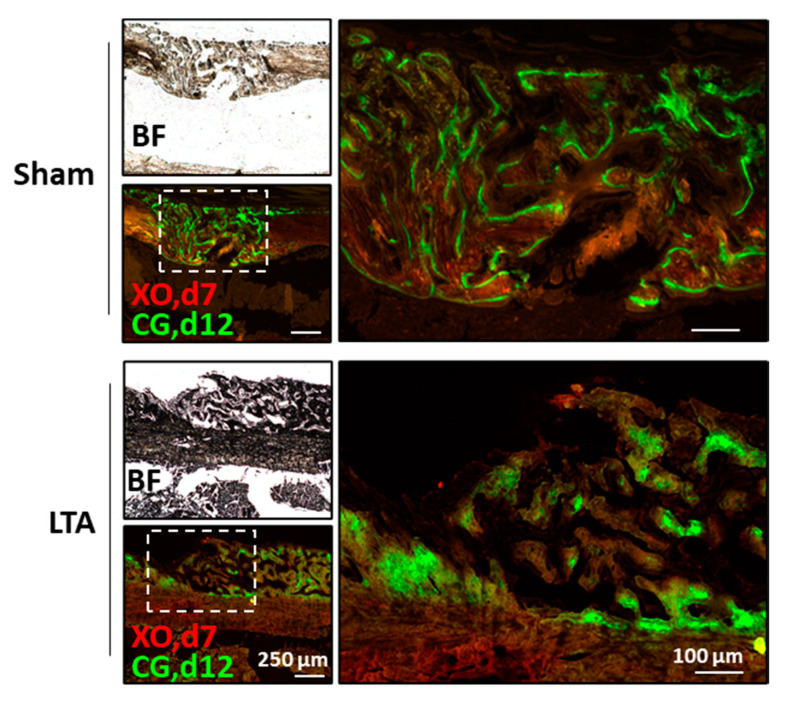
Lipoteichoic acid (LTA) showed enhanced dynamic bone-formation rates in vivo. Xylene orange (XO, Day 7) and calcein green (CG, Day 12) were used in in vivo fluorochrome-double labeling experiments to determine the onset time and location of osteogenesis.

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
