# Peer review of "Lipoteichoic Acid Accelerates Bone Healing by Enhancing Osteoblast Differentiation and Inhibiting Osteoclast Activation in a Mouse Model of Femoral Defects"

_ijms, 2020, doi:10.3390/ijms21155550_

Round 1
Reviewer 1 Report
The authors showed the osteogenic potential of lipoteichoic acid (LTA) and its ability to enhance bone healing in vivo. In vitro they showed increased alkaline phosphatase activity, calcium deposition, and osteopontin expression in MC3T3-E1 cell line when treated with lipoteichoic acid compared to vehicle-treated cells. Furthermore, cells treated with lipoteichoic acid had increased expression of MCP-1, CXCL1, cystatin C, GDF15, endostatin, and NGAL which might be a partial reason for LTA-related bone healing. Osteoclasts activity was inhibited by LTA treatment in vitro. Data suggests LTA is a good potential therapeutic option for bone regeneration but for its clinical application further evaluation is needed. Please refer to the following comments, I believe that the manuscript would be suitable for publication with major modifications.
- In Figure 1b there is asterisk marking statistical significance on day 14. Mean values seem to be overlapping, and *P<0.001 does not seem reasonable.
- In line 116 is written there are proliferating chondrocytes, but there is no proof in the data presented of cells proliferating, same marked proliferating chondrocyte in Figure 2a.
- Presented data fits nicely, but some of the sample processing and histology image quality needs to be improved (magnified images of osterix and cathepsin K, Figure 2b). The study would be more convincing if authors would perform static morphometry and show analysis of osteoblast and osteoclasts in the defect area and not just osterix and cathepsin K staining. Or evaluating numbers of osterix and cathepsin K positive cells would be an option.
- Seems like there are more cathepsin K positive cells in the 2b image of LTA treated group which is in contrast to your in vitro data showing a decreased number of TRAP+ osteoclasts. Cathepsin K is not solely expressed in osteoclasts which might be the reason there is no difference in protein and mRNA level between the groups.
- By the images provided, LTA treated group appears to heal better, but 14 days post defect healing is still not completed and stating (in line 120) that results demonstrated endochondral ossification was complete on day 14 in the LTA group is overestimated.
- Since lipoteichoic acid is a cell wall component of gram-positive bacteria, it would add up to presented data if the authors did the staining of F4/80 to determine the role of macrophages in defect healing. However, if the immune contribution to the defect healing is out of the scope of this study, authors should comment on their role in the discussion part. Additional reasons for that are presented data of increased CXCL1 that is also a chemokine affecting neutrophil migration and MCP-1 (chemokine responsible for monocyte migration), which could influence immune cell migration to defect site.
- CXCL1 attracts osteoclast precursors which could contribute to stronger cathepsin K signal 14 days post defect as seen in Figure 2b, but to evaluate LTA effect on osteoclasts more detailed analysis is needed.
- With the increased expression of MCP-1, cystatin C, GDF15, endostatin, and NGAL and their known effect on osteoblasts, evaluation of osterix positive cells within the defect should be performed to determine in vivo effect of LTA (look at previous points).
- Images of double labeling (figure 5) show a fully connected bone bridge in LTA group with callus tissue above the corticated bone, which is not the case in other images at the same time point. Cortical bone (endosteal site) bellow callus tissue has no fluorescent labels raising the question if the section presents part of the defect area.
Author Response
Please see the the attachment.

Reviewer 2 Report
Despite the extensive work in this manuscript. However, it contains a couple of minor moments.
Some of Figure legends contain a result and need to be rewritten in results.
minor comments
- please check the style in vivo(text) or in vivo(ex Figure 3);Authors need to confirm the ijmc policy (Do NOT Italicize the following: Greek/Latin expressions and non-common foreign words)
- Author missed PI3K Abbreviations.; line66
- Figure 1 legend inculed results. Line107-108; Treatment with LTA~~~ indices. This sentence should be in results section.
- Please checked p value in Fig 1 (b). Is it right?
- Fig 2 results are very important for this study. Authors need to explain more detailly. And Figure 2b must demonstrate an intense signal (Especially cathepsin K signal cannot be distiguished well.)
- The OS (full name) should be described in Figure 3. Also you need to explain "Materials and Methods".
- Figure 5 legend inculed results. Line184-1187; This sentence should be in results section.
Round 2
Reviewer 1 Report
The authors have provided some additional data, however some point are not fully explained:
- In Fig. 2a there is not red safranin O staining within the defect area, and arrow is highlighting proliferating chondrocyte. Would be careful marking them as proliferating chondrocytes.
- Image 2b still seems to be low quality.
- In the text is written that endochondral ossification was complete on day 14 after the introduction of femoral defect in the LTA (Line 136) which needs to be reformulated since images shows woven bone 14 days post defect.
- Without any quantitative data it is hard to determine if LTA really is enhancing dynamic bone-formation rates in vivo
- Osterix/DAPI and Cathepsin K/DAPI staining quantification presented in the Authors response, if the n>1 should be included in the paper results or as supplementary data.
I feel that the authors have responded to the reviewers’ queries mostly in a satisfactory manner and after additional corrections, manuscript should be accepted for publication.
